# Biofabrication of a Tubular Model of Human Urothelial Mucosa Using Human Wharton Jelly Mesenchymal Stromal Cells

**DOI:** 10.3390/polym13101568

**Published:** 2021-05-13

**Authors:** Ingrid Garzón, Boris Damián Jaimes-Parra, Manrique Pascual-Geler, José Manuel Cózar, María del Carmen Sánchez-Quevedo, María Auxiliadora Mosquera-Pacheco, Indalecio Sánchez-Montesinos, Ricardo Fernández-Valadés, Fernando Campos, Miguel Alaminos

**Affiliations:** 1Tissue Engineering Group, Department of Histology, Faculty of Medicine, University of Granada, 18016 Granada, Spain; igarzon@ugr.es (I.G.); bjaimes821@unab.edu.co (B.D.J.-P.); mcsanchez@ugr.es (M.d.C.S.-Q.); malaminos@ugr.es (M.A.); 2Instituto de Investigación Biosanitaria ibs.GRANADA, 18012 Granada, Spain; cozarjm@yahoo.es (J.M.C.); ismg@ugr.es (I.S.-M.); rfdezvalades@me.com (R.F.-V.); 3Department of Histology, Faculty of Health Sciences, University Autónoma de Bucaramanga, 680003 Santander, Colombia; 4Division of Urology, University Hospital Virgen de las Nieves, 18014 Granada, Spain; manriquepascual@hotmail.com; 5Division of Gastroenterology, Julio Hooker Digest Center, Cali 760036, Colombia; mauxymosqueramd@hotmail.com; 6Department of Human Anatomy and Embryology, University of Granada, 18016 Granada, Spain; 7Division of Pediatric Surgery, University Hospital Virgen de las Nieves, 18014 Granada, Spain

**Keywords:** urothelial mucosa, human Wharton jelly mesenchymal stromal cells, biofabrication

## Abstract

Several models of bioartificial human urothelial mucosa (UM) have been described recently. In this study, we generated novel tubularized UM substitutes using alternative sources of cells. Nanostructured fibrin–agarose biomaterials containing fibroblasts isolated from the human ureter were used as stroma substitutes. Then, human Wharton jelly mesenchymal stromal cells (HWJSC) were used to generate an epithelial-like layer on top. Three differentiation media were used for 7 and 14 days. Results showed that the biofabrication methods used here succeeded in generating a tubular structure consisting of a stromal substitute with a stratified epithelial-like layer on top, especially using a medium containing epithelial growth and differentiation factors (EM), although differentiation was not complete. At the functional level, UM substitutes were able to synthesize collagen fibers, proteoglycans and glycosaminoglycans, although the levels of control UM were not reached ex vivo. Epithelial differentiation was partially achieved, especially with EM after 14 days of development, with expression of keratins 7, 8, and 13 and pancytokeratin, desmoplakin, tight-junction protein-1, and uroplakin 2, although at lower levels than controls. These results confirm the partial urothelial differentiative potential of HWJSC and suggest that the biofabrication methods explored here were able to generate a potential substitute of the human UM for future clinical use.

## 1. Introduction

The human urinary tract is exposed to numerous diseases and conditions, including congenital malformations and acquired disorders (trauma, cancer, infections, etc.) [1]. Surgical reconstruction is challenging, and it often requires the use of autografts obtained from the skin, intestine, and other nonurological organs [2]. Although these techniques can improve the quality of life of the patients [3], the different anatomy, histological structure, and function of these organs as compared to the urinary system are often associated with important complications and drawbacks for the patient [2]. Therefore, the development of novel more effective therapeutic alternatives is needed.

Histologically, the internal face of the urinary tract is covered by a urothelial mucosa (UM) that protects these organs from the hostile environment of urine [4]. The main function of the urothelium is protection of the underlying structures by preventing water infiltration. To achieve this function, urothelial cells are tightly compacted by the presence of numerous intercellular junctions and by the formation of urothelial plaques formed by a special type of proteins (uroplakins) at the apical membrane of terminally differentiated umbrella cells [5]. Replacement of the UM by nonurological tissues rarely replace the entire function of the original organ due to the important histological differences between both structures.

In this regard, advancements in tissue engineering have allowed for the generation of bioartificial substitutes able to restore, replace, or improve the function of damaged tissues and organs [6]. The possibility of generating a specific urinary tract organ in the laboratory by combining functional cells and biocompatible biomaterials opens the door to the creation of tissue substitutes capable of resembling the native structure and function without the drawbacks as associated with the use of nonurological tissue grafts. Several bioartificial substitutes of the urinary system have been described in the literature [7], including several models of the human urethra [8,9], bladder [10,11,12], and ureter [13]. Although some of these models showed good preclinical and clinical results, the ideal substitute of the human UM has not been described to date. One of the main challenges in tissue engineering is finding an appropriate cell source able to provide large cell quantities [7], since many cell types, especially human epithelial cells, are difficult to culture and show very low proliferation rates [14,15,16]. One of the possible alternative cell sources for the generation of human cell cultures is the umbilical cord Wharton’s jelly [17]. human Wharton jelly mesenchymal stromal cells (HWJSC) are immunoprivileged and can be easily isolated from small fragments of the human umbilical cord, showing high proliferation and differentiation potential [17]. HWJSC have been efficiently differentiated from several cell types, including epithelial cell lineages such as skin and oral mucosa keratinocytes [18,19], and corneal epithelial cells [20]. Although very few works focused on the urothelial differentiation potential of HWJSC, interesting preliminary reports suggest that these cells might be able to be differentiated into urothelium-like cells both ex vivo [21] and in vivo [22], at least, partially.

In this work, we evaluated the capability of HWJSC to be differentiated into the urothelial cell phenotype using different induction factors in a heterotypical model of the human UM generated by tissue engineering. In addition, the bioartificial UM was subjected to novel biofabrication methods, allowing us to generate a tubular organ that might resemble the spatial structure of the urinary tract organs.

## 2. Materials and Methods

### 2.1. Establishment of Primary Cell Cultures

Small fragments of the distal human ureter were obtained from patients subjected to bladder removal surgery (cystectomy) at the Urology Department of the University Hospital Virgen de las Nieves of Granada, Spain. To generate urothelial primary cell cultures, biopsies were washed in PBS and fragmented in small pieces that were placed on the surface of a Petri dish (Sarstedt, Nümbrecht, Germany) with the epithelial face upside down to favor urothelial expansion on the culture surface using the explant technique. Next, 30 min later, epithelial culture medium (EM) was carefully added. This medium consisted of a mixture of HAM-F12 (150 mL), DMEM (300 mL), and fetal bovine serum (50 mL) supplemented with penicillin/streptomycin (50 IU/mL), adenine (24 μg/mL), insulin (5 μg/mL), triiodothyronine (1.3 ng/mL), hydrocortisone (0.4 μg/mL), and epidermal growth factor (EGF) (10 ng/mL) (all of them, from Millipore Sigma—Merck Life Science, St. Louis, MO, USA). To obtain primary cell cultures of urinary tract stromal cells, biopsies were enzymatically digested in a 2 mg/mL solution of *Clostridium histolyticum* collagenase I (Gibco—Thermo Fisher Scientific, Waltham, MA, USA) at 37 °C. Biopsies were incubated in this solution and examined every 30 min until the tissue was completely digested (typically, after 1–2 h, and up to 6 h for larger biopsies); this solution was centrifuged to harvest isolated cells, and cells were cultured in a basic culture medium consisting of Dulbecco’s modified Eagle’s medium (DMEM) supplemented with 10% fetal bovine serum and 1% antibiotics (all of them, from Millipore Sigma—Merck Life Science, St. Louis, MO, USA).

To generate HWJSC cultures, fragments of the human umbilical cord were obtained from newborns delivered by cesarean section, and small pieces of the jelly tissue allocated among the blood vessels were obtained. These pieces were then digested in a mixture of collagenase I and a solution containing 0.5 g/L trypsin and 0.2 g/L EDTA (Gibco—Thermo Fisher Scientific, Waltham, MA, USA). Cells were collected by centrifugation and cultured in HWJSC culture medium (WM) consisting in commercial Amniomax-C100 culture medium (Gibco—Thermo Fisher Scientific, Waltham, MA, USA). All cell types were kept at 37 °C in a humidified incubator (Esco Lifesciences Group Ltd, Singapore, Singapore) with 5% CO_2_ using standard cell culture conditions. HWJSC were characterized for CD90, CD45, CD105, and CD73 markers by flow cytometry (BD Biosciences, Franklin Lakes, NJ, USA) before use.

This study was approved by the Research and Ethics Committee in Biomedical Research of Granada (project code 32140077, approval date 26 January 2015). Samples were provided by the Biobank of the Public Health System of Andalusia, with the informed consent of all donors.

### 2.2. Generation of Heterotypical Substitutes of the Human Urothelial Mucosa (UM)

To generate heterotypical substitutes of the human urothelial mucosa (UM) by tissue engineering, we first fabricated a biological substitute of the suburothelial connective tissue (stromal layer), and epithelial cells were then subcultured on top. To do so, 500,000 stromal cells were trypsinized, resuspended in 2 mL of DMEM, and mixed with 24.3 mL of human blood plasma obtained from healthy donors, 200 μL of tranexamic acid (as an antifibrinolytic agent), 1.5 mL of 2% agarose melted in PBS, and 2 mL of 1% Cl_2_Ca solution. This mixture was aliquoted in 6-well plates and allowed to jellify at 37 °C in a cell incubator. Twenty-four h later, 250,000 HWJSC were resuspended in 500 µL of DMEM and seeded on top of the jellified stromal substitute (all reagents purchased from Millipore Sigma—Merck Life Science, St. Louis, MO, USA).

Once generated, bioartificial tissues were cultured with three different culture media, and the following groups of samples were established:1)UM-WM samples, corresponding to UM substitutes cultured in HWJSC basal medium (WM).2)UM-EM samples, corresponding to UM substitutes cultured in epithelial differentiation medium (EM).3)UM-PM samples, corresponding to UM substitutes cultured in preconditioned medium (PM).

To generate the PM medium, we first harvested the supernatant culture medium used in the urothelial primary cell cultures, after 3 days of culture. Then, this medium was filtered using a 0.22 µm syringe filter and mixed with a similar volume of fresh EM. All culture media were renewed every 3 days, and UM substitutes were kept in culture for 7 and 14 days.

After this time, UM substitutes were extracted from the culture plates and subjected to plastic compression nanostructuration as previously described [23]. For this, we first applied a sterile surgical dressing (Mölnlycke Health Care, Madrid, Spain) with a pore size of 1 mm on the epithelial layer of the UM to protect this layer from excessive damage during the procedure. Then, two pieces of sterile 3 MM absorbent paper (Millipore Sigma—Merck Life Science, St. Louis, MO, USA) were placed on the nanostructuration surface, and a sterile nylon filter with a pore size of 0.22 μm (Millipore Sigma—Merck Life Science, St. Louis, MO, USA) was placed on top, and the UM substitute was carefully positioned above. A second nylon filter and two additional pieces of 3 MM absorbent paper were placed on the UM substitute, and a piece of glass weighing 500 g was carefully deposited on top of the system to apply a controlled pressure for 3 min. After this time, nanostructured UM substitutes were recovered and kept in PBS to prevent from excessive drying. This nanostructuration protocol was previously optimized by our research group for fibrin–agarose biomaterials [24]. After nanostructuration, each bioartificial tissue was tubularized to mimic the 3D structure of native urinary tract organs. To accomplish that, the nanostructured tissue was cut in a rectangular shape of approximately 2 cm × 1 cm, and a silicone tube guide was carefully placed on the epithelial surface of the construct. Then, both ends of the UM were sutured using 8/0 silk suture material in order to obtain a tubular structure (Figure 1A), and the tube was finally removed.

### 2.3. Histology, Immunofluorescence, and Immunohistochemistry

Histological analysis of control human native UM and bioengineered UM was performed by light and scanning electron microscopy. For light microscopy, tissues were fixed in 4% formaldehyde, dehydrated and embedded in paraffin using routine histological protocols (all reagents purchased from PanReac AppliChem, Barcelona, Spain). Four μm thick sections were obtained on glass slides, which were subjected to dewaxing in xylene and progressive rehydration in ethanol series. The general structure of each sample was analyzed by hematoxylin-eosin (H&E) staining. Briefly, samples were incubated for 3 min in hematoxylin (PanReac AppliChem, Barcelona, Spain), rinsed in tap water for 5 min, and furtherly stained with eosin for 1 min (PanReac AppliChem), followed by ethanol dehydration, and mounted using glass coverslips. For scanning electron microscopy (SEM), samples were fixed in 2.5% glutaraldehyde, dehydrated with acetone, and critical-point dried. Samples were then mounted on aluminum stubs, sputter-coated with gold-palladium and examined in a Quanta 200 scanning electron microscope (FEI, Eindhoven, The Netherlands). In addition, histological analysis was performed on samples fixed in 2.5% glutaraldehyde, dehydrated in acetone, embedded in epoxy resin and cut in semithin 0.5 µm sections with a Leica ultracut UCT ultramicrotome (Leica Microsystems GmbH, Wetzla, Germany) which were stained with Toluidine blue (PanReac AppliChem, Barcelona, Spain).

Evaluation of the main fibrillar and nonfibrillar components of the extracellular matrix (ECM) of the stromal substitute was carried out by histochemistry as previously described [25,26]. Identification of collagen fibers was performed by picrosirius red [27], whereas reticular fibers were stained with the Gomori’s reticulin histochemical method. Assessment of ECM proteoglycans were achieved with alcian blue (PanReac AppliChem, Barcelona, Spain), and glycoproteins were identified with the periodic acid Schiff (PAS) histochemical method.

Characterization of the epithelial layer of control and bioengineered UM was carried out by immunofluorescence and immunohistochemistry for specific markers of the human urothelium. Specifically, several cytokeratins (pancytokeratin and keratins 13, 7 and 8) and the cell–cell junction proteins desmoplakin (DSP) and tight-junction protein-1 (TJP1 or ZO1) were identified by immunofluorescence, whereas two markers of terminal urothelial differentiation (uroplakins 2 and 3) were detected by immunohistochemistry.

For immunohistochemistry, tissue sections were treated with H_2_O_2_ to inactivate endogenous peroxidases, and antigen retrieval was carried out with citrate or EDTA buffer. Samples were washed and incubated in CAS-Block™ histochemical reagent (Thermo Fisher Scientific) to block nonspecific antigens, followed by incubation with the primary antibody: rabbit antiuroplakin 2 (ref. HPA043312, Millipore Sigma—Merck Life Science), dilution 1:60; mouse antiuroplakin 3 (ref. ab78197, Abcam, Cambridge, UK), prediluted. After washing and incubating with peroxidase labeled secondary antibodies (Vector Laboratories, Burlingame, CA), a diaminobenzidine (DAB) development kit (Vector Laboratories) was used to trigger the immunohistochemical reaction. Samples were then briefly contrasted with Harris hematoxylin and mounted using glass coverslips. For immunofluorescence, tissue sections were subjected to antigen retrieval as described for immunohistochemistry. Then, samples were washed in PBS and incubated with the primary antibody: mouse anti-pancytokeratin (ref. C2931, Millipore Sigma—Merck Life Science), dilution 1:100; mouse anti-cytokeratin 13 (ref. C0791, Millipore Sigma—Merck Life Science), dilution 1:400; mouse anti-cytokeratin 7 (ref. MAD-001004QD3, Master Diagnóstica, Granada, Spain), prediluted; rabbit anti-cytokeratin 8 (ref. MAD-000693QD3, Master Diagnóstica), prediluted; mouse anti-desmoplakin (ref. AM09122SU-N, Acris, Herford, Germany), prediluted; rabbit antitight-junction protein 1 (ZO1) (ref. HPA001637, Millipore Sigma—Merck Life Science), and dilution 1:75. After washing, an antimouse secondary antibody labeled with FITC or an antirabbit secondary antibody labeled with Cy3 was applied. Finally, nuclei were counterstained using DAPI, and samples were covered with coverslips.

In all cases, images were obtained with a Nikon Eclipse 90i light microscope (Nikon Corp., Tokyo, Japan) using the same conditions for each staining method—exposure, contrast, brightness, white balance, background, etc.—with the Nikon NISElements software. Then, for the immunofluorescence analyses, in which tissue distribution is more important than intensity, results were semiquantitatively scored by three independent histologists using a previously reported scale [26,28]. For each immunofluorescence marker, fluorescence signal was evaluated in each image as strong positive signal (+++), positive signal (++), mild signal (+), or negative (−), according to the general distribution of the signal in each sample. For histochemistry and immunohistochemistry, in which signal intensity is crucial, the staining signal was quantitatively assessed as previously reported [26]. Briefly, signal intensity was quantified in each histological image using the multipoint tool of ImageJ software (National Institutes of Health, Bethesda, MD, USA). Ten random points were selected, and the intensity was calculated by the program. Averages and standard deviations were obtained after subtracting the background white signal. Statistical comparisons between values were obtained for control samples, and each sample type was performed by the Mann–Whitney tests with the RealStatistics software (Dr. Charles Zaiontz, Purdue University, West Lafayette, IN, USA). The same test was used to compare each type of UM with the other UM corresponding to the same development time and to compare the results obtained at different times for the same type of UM.

## 3. Results

### 3.1. Histological Analysis of Heterotypical Substitutes of the Human Urothelial Mucosa (UM)

In the first place, we found that the HWJSC used in the present study fulfilled the criteria established by Dominici et al. (2006) for this type of cells [29]. Specifically, 94.2% of cultured cells were positive for CD90, 94.7% for CD105, 99.1% for CD73, and 0.45% for CD45.

Then, histological analysis of the bioengineered UM generated in this work using light and scanning electron microscopy (Figure 1B) revealed that these bioartificial tissues were formed by a central light surrounded by a dense biomaterial. Bioengineered UM retained a tubular shape that was partly comparable to the native normal ureter used as control. However, differences were found in light of the UM, which was larger in UM and was devoid of the internal folds found in control ureter.

Structural analysis at higher magnification (Figure 2) showed that human native ureter consisted of a stroma with abundant fibroblast cells immersed in a fibrillar extracellular matrix (ECM) and an epithelium showing nuclei disposed at different levels, with clear signs of cell differentiation. In turn, we found that bioengineered UM were composed by a fibrin–agarose biomaterial containing scattered elongated, spindle-shaped cells that were compatible with a stromal substitute, and a cell layer resembling an epithelium on top. Analysis of bioengineered UM cultured with WM for 7 days (UM-WM) showed 2–3 epithelial-like cell strata on the fibrin–agarose stromal substitute, and cells did not show any signs of urothelial differentiation. After 14 days, the number of cells on top of the stromal substitute increased to 5–7 layers. However, when the UM were cultured with EM (UM-EM), we observed the presence of an epithelial-like cell layer with around 10 cell strata after 7 and 14 days of ex vivo development. In UM-EM at day 14, cells in the epithelial layer showed incipient differentiation signs, with a flattened morphology in the most superficial strata and rounded or prismatic shape in the basal strata. Finally, UM cultured with PM (UM-PM) revealed the presence of a thin epithelial-like cell layer with 2–3 cell strata after 7 and 14 days of showing no signs of urothelial differentiation.

### 3.2. Histochemical Characterization of the UM Stromal Substitute

To characterize the stromal layer of controls and bioengineered UM, we first analyzed and quantified the presence of fibrillar components of the ECM by picrosirius red and Gomori’s reticulin histochemical methods (Table 1). In this regard, we found a strongly positive picrosirius red signal (Figure 3) in the stromal layer of control human UM, suggesting that collagen fibers were very abundant at this level. By contrast, all bioengineered UM showed significantly lower picrosirius red staining intensity, especially in samples corresponding to 7 days of follow-up, which were significantly lower than 14 days samples for UM-WM and UM-EM. Interestingly, the bioartificial samples showing higher picrosirius red staining intensity were UM-WM cultured for 14 days, although signal did not reach the levels found in controls. When reticular fibers were assessed using Gomori’s reticulin histochemistry (Figure 3 and Table 1), we found that these fibers were present in control native UM at significantly higher levels than bioengineered UM tissues, with nonsignificant differences among UM samples.

Then, stromal characterization was performed by analyzing nonfibrillar components of the ECM by alcian blue and PAS histochemical methods. For alcian blue (Figure 4 and Table 1), we found that control native UM showed strong positive signal in the connective tissue and were negative in the urothelial layer. However, bioengineered UM samples showed very low signal in the stromal layer (significantly lower than control samples) and showed different degrees of positivity in the epithelial-like layer. UM-WM corresponding to 7 days of development showed significantly higher intensity than UM-EM and UM-PM at the same time of development. At day 14, UM-WM samples showed significantly more intense signal than UM-PM samples at 14 days but were statistically comparable to UM-EM tissues at 14 days. For the ECM glycoproteins identified by PAS (Figure 4 and Table 1), we found positive expression of these components at the basal lamina of control native UM and in UM-WM and UM-EM samples corresponding to 7 and 14 days of development, with nonsignificant differences among all these samples. By contrast, UM-PM tissues at 7 days were significantly lower than controls and UM-EM at 7 days, and UM-PM at 14 days were significantly lower than controls, UM-EM at 14 days and UM-WM at 14 days.

### 3.3. Characterization of the UM Epithelial-Like Layer by Immunofluorescence and Immunohistochemistry

The expression analysis of cytokeratins using pancytokeratin immunofluorescence revealed a strong positive signal in the urothelial layer of native human UM (Figure 5 and Table 2). When bioartificial UM were analyzed, we found that bioartificial tissues cultured in WM and PM media expressed positive pancytokeratin signal, whereas UM-EM tissues showed mild signal at the epithelial-like layer, with no differences between times. Regarding cytokeratins 13 (Figure 5) and 7 (Figure 6), controls showed strong immunofluorescence signals, and UM-EM at day 14 presented mild signal at the epithelial-like layer. The rest of samples were negative for these two cytokeratins. Analysis of cytokeratin 8 showed strong positivity in controls, and mild or positive signal in all UM-WM and UM-PM samples, and at day 14 UM-EM bioartificial tissues (Figure 6).

Characterization of the epithelial-like cell layer was also carried out by analyzing the expression of two relevant cell–cell junction proteins (desmoplakin and tight-junction protein-1). Results showed that the native urothelium was strongly positive for both markers (Figure 7 and Table 2), confirming that well-developed intercellular junctions are present in this epithelium. Immunofluorescence detection of desmoplakin in bioartificial UM showed that tissues generated in the laboratory expressed this protein at a certain level, although the signal found in controls was not reached. The most intense expression corresponded to UM-PM at day 14 (positive signal), and the rest of the samples showed mildly positive intensity. In turn, tight-junction protein-1 only showed a positive signal in UM cultured with EM medium, especially in samples corresponding to 14 days of development, although samples kept ex vivo for 7 days were also positive.

To furtherly characterize the epithelial-like layer of all samples included in the present study, we analyzed the expression of uroplakins 2 and 3 by immunohistochemistry (Figure 8 and Table 1). As expected, we found a strong positive signal of both proteins at the most superficial layer of control human urothelium, with statistically significant differences with all types of bioengineered UM. For uroplakin 2, results showed that all bioartificial tissues were able to express certain amounts of this protein, although at lower levels than controls. Unlike controls, expression was homogeneous in all cells found at the epithelial-like cell layer of bioengineered UM. Although our statistical analysis revealed nonstatistically significant differences among groups, 14 days samples tended to show higher expression than 7 days samples. Uroplakin 3 signal was negative in all bioengineered UM samples.

## 4. Discussion

Several models of bioartificial substitutes of the human urinary tract have been described to the date [7,10,11,12,13]. However, the drawbacks and complications associated with most of these models make necessary the development of improved UM substitutes. Among the different biomaterials previously used in tissue engineering, fibrin–agarose offers numerous advantages such as low price, easy manufacturing, high availability, and biocompatibility [19]. In fact, fibrin–agarose biomaterials allowed the successful generation of different biological substitutes of the human cornea, skin, oral mucosa, and other tissues [19,30,31], including a model of the human urinary bladder mucosa showing promising results [12]. The positive results obtained at the preclinical level allowed us to obtain approval by the Spanish National Medicines Agency (Agencia Española de Medicamentos y Productos Sanitarios-AEMPS) for clinical use in patients with severe skin burns and corneal ulcers [31,32,33].

An important concern in tissue engineering of urinary tract organs is the development of biofabrication methods allowing the efficient generation of functional tubular structures. While flat tissues and organs such as the skin can be categorized as low complexity, tubular organ structures are at the next level of complexity [7]. In fact, we have previously described a flat model of the human UM based on fibrin–agarose hydrogels that showed positive ex vivo results but did not allow tubulization due to the poor biomechanical properties of this biomaterial [12]. The combination of fibrin–agarose and nanostructuration methods allowed us to generate flat structures and solid cylinders, but this is the first description of a hollow tubular structure made with fibrin–agarose biomaterials. The three-dimensional shape of this structure was partially analogue to some organs of the urinary tract such as the human ureter or urethra, with an internal epithelial-like layer surrounding a light and an external stromal substitute [4]. Previous studies confirmed the biomechanical resistance of artificial tissues based on fibrin–agarose scaffolds [23,24,34,35]. However, the biomechanical behavior of the new UM substitutes should be tested in future studies.

One of the advantages of this biofabrication method is the possibility of generating hollow tubes of different light areas by using a tube guide of different thickness. Although this organ should still be improved by the addition of a muscular and an adventitia layers, its structure is a significant advance as compared to previously available flat models [12]. However, this biofabrication method should still be improved to overcome the problems associated with biomaterial suturing and to improve integrity of the tubular structure, for example, through the use of biological glue or other types of biomaterials.

Apart from the 3D morphology, one of the main findings of the present work is the possibility of using HWJSC as an alternative source of cells able to generate a substitute of the human UM. Most of the available bioartificial UM models use urothelial cells obtained from UM biopsies, which may be prejudicial for the patient and limit the possibility of obtaining large quantities of cells in culture. In contrast, HWJSC previously demonstrated high differentiation potential into several cell phenotypes, including different types of epithelial cells [17,20], although it is not clear if these cells can be able to differentiate into different cell lineages or if the process is mediated by cell fusion [36,37]. HWJSC have several advantages compared with other cell types, including accessibility, differentiation potential, and high rate of proliferation, and these cells have low immunological rejection rates when allogeneic cells are implanted in a patient. For these reasons, HWJSC have been widely used in tissue engineering [17,38]. Despite their promising properties, HWJSC have been very scarcely used in tissue engineering of the urinary system, although available results suggest that these cells may have potential usefulness [21,22].

In the present work, we found that HWJSC were able to differentiate, at least partially, toward the urothelial cell lineage, although the culture conditions significantly influenced the differentiation efficiency. To achieve urothelial differentiation, a combined approach was used. In the first place, a 3D biomimetic substitute of the human UM was generated with stromal cells isolated from the suburothelial stroma. The use of 3D systems allows epithelial–mesenchymal interaction and differentiation induction driven by paracrine factors synthesized and secreted by tissue specific stromal cells immersed in the biomaterial [39,40]. The use of these systems demonstrated to be effective in inducing ex vivo differentiation into different tissue types [18,41,42], although we only found a partial effect.

In the second place, specific inductive media were used. The influence of differentiation media has been demonstrated in different settings [17], although very little information related to the specific composition of urothelial inductive media is currently available. One of the previous reports assessing this issue [21] suggested that a combination of EGF and conditioned medium derived from primary cultures of urothelial cells could induce HWJSC differentiation into the urothelial lineage in bidimensional cell cultures, but no consensus exists. For this reason, in the present work we evaluated three types of differentiation media to determine which of them could more efficiently induce urothelial differentiation: the basal medium used for HWJSC culture, a culture medium containing different epithelial differentiation factors such as EGF, and a preconditioned medium generated by human cultured urothelial cells. Future studies should determine the exact differentiative efficiency of each of these media on different cell settings to properly evaluate the effects of each media on epithelial differentiation.

In general, our results suggest that the combined approach used in this work could exert a positive effect on urothelial induction of HWJSC, although terminal urothelial differentiation was not achieved in any of the experimental conditions. Although some preliminary signs of urothelial differentiation were found, most induced HWJSC significantly differed from control urothelium from a morphological and functional level. Histologically, we found that cells corresponding to the UM-WM and UM-PM groups reached very few epithelial-like strata and scarce signs of epithelial differentiation. In contrast, the use of EM resulted in numerous cell strata with incipient signs of differentiation, suggesting that the use of soluble inductive factors in the culture medium is able to trigger epithelial differentiation of HWSJC, as previously demonstrated for other bioartificial tissue models [19,20,30] and for the human bladder [21].

Analysis of the stromal substitute using different histochemical methods revealed several differences among the study groups and confirmed the crucial importance of the epithelial–mesenchymal interaction in cell and tissue differentiation [39,40]. In general, the use of EM was associated with the synthesis of certain amount of collagen and reticular fibers, proteoglycans, and glycoproteins in the bioartificial UM, although at lower level than controls, except for glycoproteins. Although future studies should confirm these preliminary results, our data support the higher efficiency of EM over WM and PM as an inductor of the suburothelial stromal layer. Interestingly, UM-WM showed the highest levels of proteoglycans at the stromal substitute. This could be explained by the fact that WM is the basal medium generally used for HWJSC culture, and cells kept in this medium typically express high amounts of these ECM components [43]. Although each culture medium was able to induce certain level of stromal differentiation, it is noteworthy that none of the experimental conditions succeeded in generating a fully differentiated stroma substitute comparable to the native structure.

Moreover, the immunofluorescence and immunohistochemistry analysis of the epithelial-like layer of bioengineered UM confirmed the idea that HWJSC may have urothelial differentiation potential. However, as in the case of the stromal layer, differentiation was very limited at this level and varied among the different experimental conditions. One of the most important markers of epithelial function are cytokeratins [44], which participate in cell–cell adhesion and are fundamental for the barrier function of human epithelia. Our analysis revealed that HWJSC were able to express several cytokeratins in all the study groups, confirming our previous findings showing that these cells have intrinsic potential to express these epithelial markers [45]. However, we found that the type and amount of cytokeratins significantly varied among the study groups and depended on the culture media. In general, the use of EM as a differentiation medium for 14 days was the only condition allowing for certain expression of all cytokeratins expressed by the native UM (pancytokeratin and cytokeratins 13, 7, and 8). In general, cytokeratin expression was very low in the UM-WM and UM-PM groups, suggesting that the inductive potential of WM and PM could be lower as compared to EM.

Along with cytokeratins, intercellular junction proteins are key components allowing the barrier function of human urothelium [46]. Herein, we analyzed two cell–cell junction proteins that are very abundant in native UM: desmoplakin, one of the components of desmosomes, and tight-junction protein-1, one of the constituents of tight junctions. Results showed that EM was able to induce the expression of both types of proteins, especially after 14 days of development, although the levels found in controls were not reached. As the presence of tight-junction protein-1 is considered to be crucial to control urothelial permeability [47], these results probably imply that the epithelial-like layer of UM-WM and UM-PM could not function as an efficient impermeable barrier, at least, at this stage of ex vivo development. Future studies should determine the transpermeability of these bioartificial UM using functional techniques.

Finally, we analyzed the expression of two uroplakins in the developing epithelial-like layer of bioartificial UM. Uroplakins are cell-membrane proteins, which form urothelial plaques at the apical surface of the urothelium, and their presence is necessary for the UM to exert its functions [5]. In our case, expression of uroplakins 2 and 3 was very low in all UM generated in the laboratory, especially in the case of uroplakin 3. Although the differences were not statistically significant, a trend was found of a higher expression of uroplakin 2 in 14 days samples as compared to earlier tissues, which allows us to hypothesize that longer development times could result in improved differentiation rates. All these results are in agreement with those found for other epithelial markers and suggest that the differentiation level of UM samples kept ex vivo is positive, although incipient.

The reasons why the EM inductive medium combined with the 3D induction system appears to be more effective than WM and PM media should be determined in future studies. However, we may hypothesize that the hormones and growth factors supplied by EM, including EGF, could be the responsible for the partial urothelial differentiation achieved with this medium. In fact, EM is routinely used by our research group to successfully culture different types of epithelial cells [48,49,50]. The fact that PM medium did not seem to be as efficient as EM is intriguing, since PM should contain most of the factors found in EM, along with a series of molecules liberated by human urothelial cells, and previous reports showed that PM was able to induce urothelial differentiation of HWJSC bidimensional cell cultures on culture flasks [21]. Probably, the use of 3D models of the human UM and the different composition and EGF concentration used in our work could partially explain the different results obtained in the present study. Another possibility is that the toxic molecules and waste products released to PM by urothelial cultures could be deleterious for differentiating HWJSC, and modifications in the formulation and elaboration procedure of PM could improve its inductive potential.

All these results confirm previous observations demonstrating that bioartificial tissue substitutes kept in culture only display partial levels of cell and tissue differentiation, and that in vivo grafting is often able to induce terminal maturation and differentiation [51]. Most likely, the inductive factors used ex vivo are not sufficient for achieving complete differentiation, and a combination of additional factors found in vivo such as biomechanical clues, vascularization, innervation, and a plethora of paracrine and endocrine molecules, are necessary for an efficient tissue differentiation [49,51]. In fact, interesting reports by Yuan et al. suggest that HWJSC can be efficiently differentiated into urothelial-like cells without the use of any inductive media, if these cells are grafted in vivo in a model of bioartificial urinary bladder in dogs [22].

The present study had several limitations. In the first place, the three differentiation media used in this work will require further characterization to demonstrate its safety and efficiency and the mechanisms associated with their inductive effect. In the second place, the bioartificial UM generated in this work should be evaluated to demonstrate tissue functionality, especially referring to urothelial barrier function, and future works should determine the barrier function of this epithelium. Another limitation is the lack of an in vivo characterization in laboratory animals. Although previous preliminary works carried out by our research group point out the good biocompatibility of the fibrin–agarose biomaterial used for different tissue engineering applications in animals [27] and in humans [31,32], in vivo characterization will definitely contribute to shedding light on the properties of the UM bioartificial tissues in terms of biocompatibility and functionality.

## 5. Conclusions

In the present work, we have been able to generate bioartificial UM able to resemble native UM in terms of 3D morphology and share partial histological and functional similarities with the native tissue. The application of the biofabrication methods described here allowed us to generate hollow tubular structures that could be clinically used to replace some of the organs of the urinary tract such as the ureter and the proximal urethra. In addition, we demonstrated that UM can be generated from a nonurological cell source obtained from the umbilical cord, since these cells have certain urothelial differentiation potential ex vivo, although differentiation was not complete and induced HWJSC that differed from control urothelium at several levels. Future in vivo experiments should determine if HWJSC can generate fully differentiated UM once grafted in vivo.

## Figures and Tables

**Figure 1 polymers-13-01568-f001:**
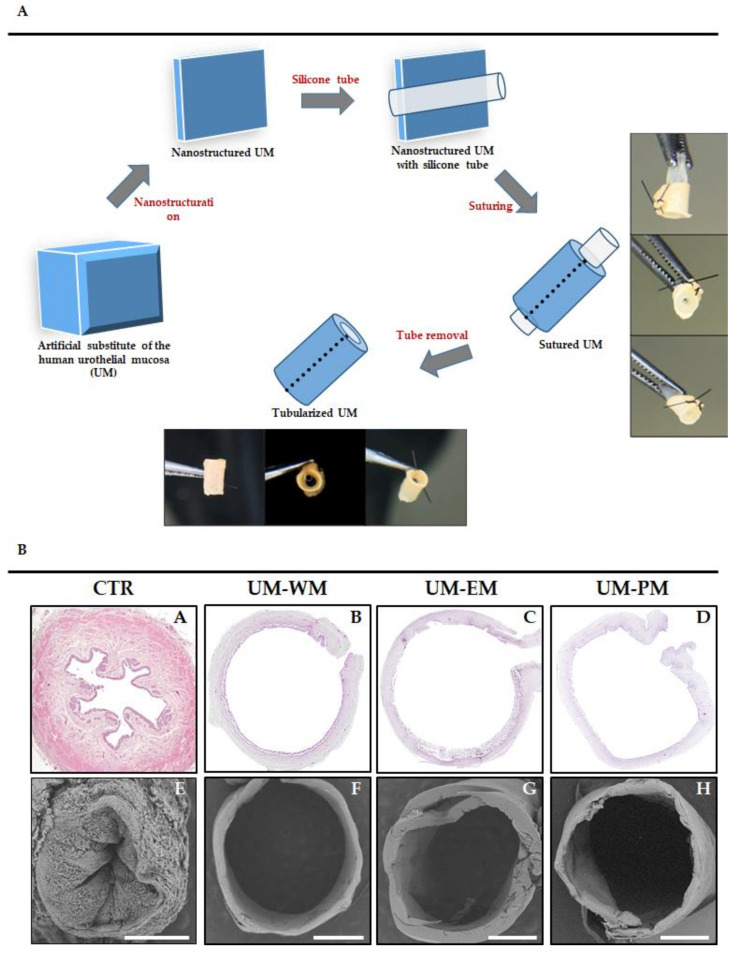
(**A**) Procedure used for the generation of tubularized heterotypical substitutes of the human urothelial mucosa (UM). (**B**) Low-magnification histological analysis of control human UM (CTR) and tubularized heterotypical substitutes of the human UM cultured with three different media (WM, EM and PM) and stained with hematoxylin-eosin (**A**–**D**) or analyzed with scanning electron microscopy (**E**–**H**). Scale bars: 1 mm.

**Figure 2 polymers-13-01568-f002:**
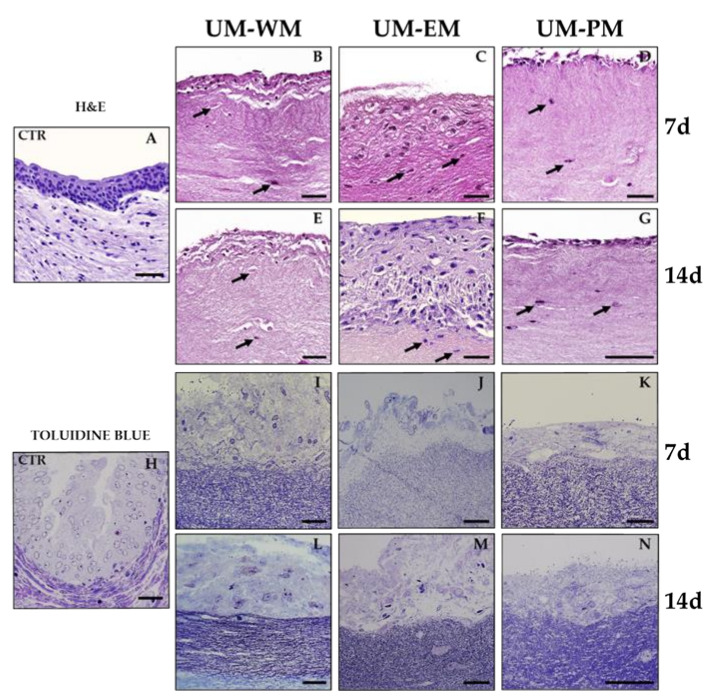
Histological analysis of control human UM (CTR) and tubularized heterotypical substi-tutes of the human UM cultured with three different media (WM, EM, and PM) for 7 and 14 days and stained with hematoxylin-eosin (**A**–**G**) and semithin sections with toluidine blue (**H**–**N**). Illustrative cells have been highlighted with arrows in the stroma of bioartificial UM tissues stained with hematoxylin-eosin. Scale bars: 50 µm.

**Figure 3 polymers-13-01568-f003:**
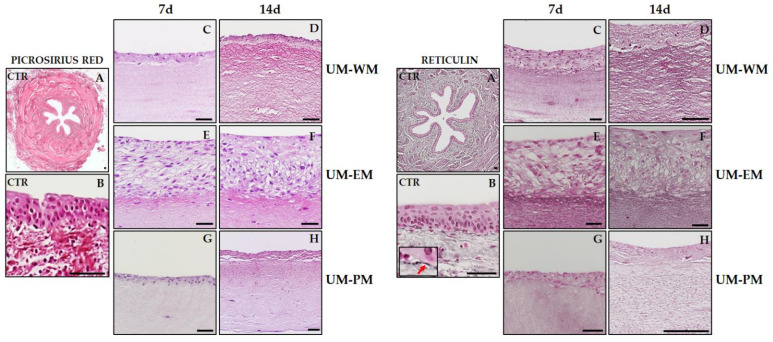
Picrosirius red and Gomori’s reticulin histochemical analysis of control human UM (CTR) and tubularized heterotypical substitutes of the human UM cultured with three different media (WM, EM, and PM) for 7 and 14 days. Illustrative reticular fibers are highlighted with a red arrow in the high-magnification insert in reticulin panel (**B**). Scale bars: 500 µm in (**A**) and 50 µm in (**B**–**H**).

**Figure 4 polymers-13-01568-f004:**
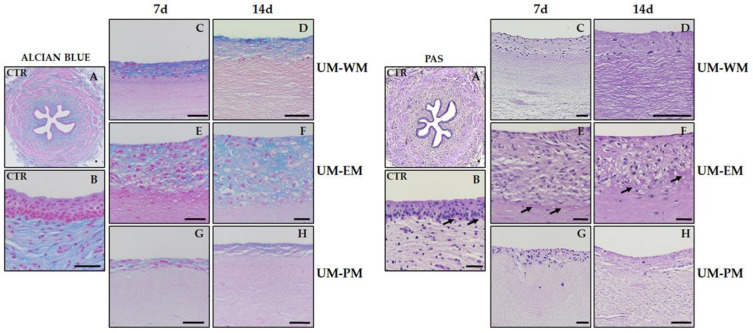
Alcian blue and PAS histochemical analysis of control human UM (CTR) and tubularized heterotypical sub-stitutes of the human UM cultured with three different media (WM, EM, and PM) for 7 and 14 days. Illustrative areas showing positive signal are highlighted with black arrows in the PAS panel. Scale bars: 500 µm in (**A**) and 50 µm in (**B**–**H**).

**Figure 5 polymers-13-01568-f005:**
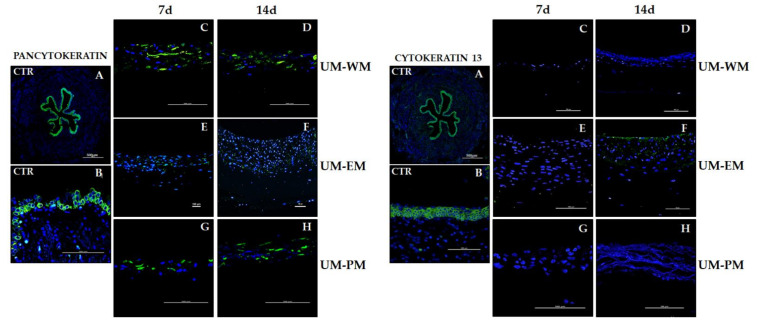
Immunofluorescence analysis of pancytokeratin and cytokeratin 13 in control human UM (CTR) and tubularized heterotypical substitutes of the human UM cultured with three different media (WM, EM, and PM) for 7 and 14 days. Scale bars: 500 µm in (**A**) and 100 µm in (**B**–**H**) panels.

**Figure 6 polymers-13-01568-f006:**
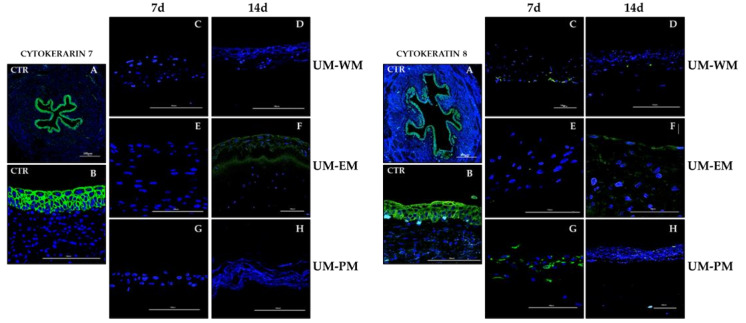
Immunofluorescence analysis of cytokeratins 7 and 8 in control human UM (CTR) and tubularized heterotypical substitutes of the human UM cultured with three different media (WM, EM, and PM) for 7 and 14 days. Scale bars: 500 µm in (**A**) and 100 µm in (**B**–**H**) panels.

**Figure 7 polymers-13-01568-f007:**
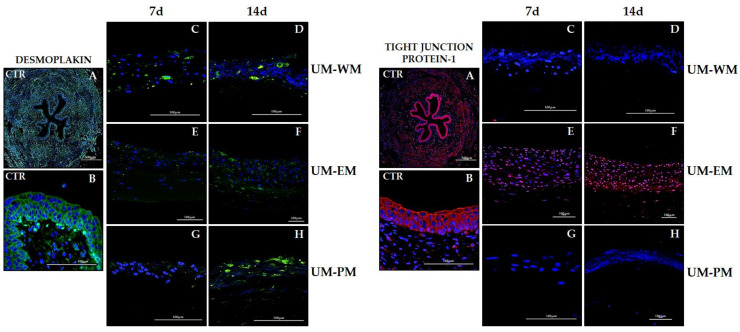
Immunofluorescence analysis of desmoplakin and tight-junction protein-1 (TJP1) in control human UM (CTR) and tubularized heterotypical substitutes of the human UM cultured with three different media (WM, EM, and PM) for 7 and 14 days. Scale bars: 500 µm in (**A**) and 100 µm in (**B**–**H**) panels.

**Figure 8 polymers-13-01568-f008:**
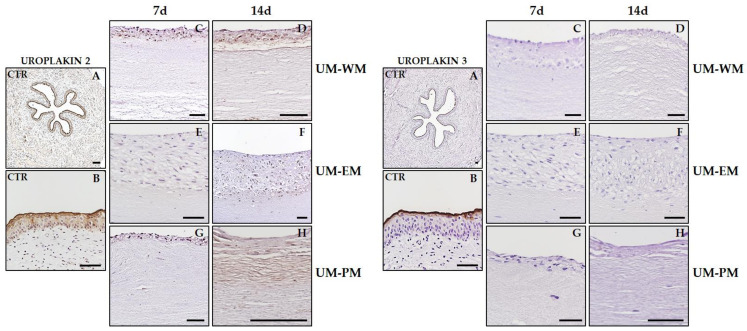
Immunohistochemical analysis of uroplakins 2 and 3 in control human UM (CTR) and tubularized heterotypical substitutes of the human UM cultured with three different media (WM, EM, and PM) for 7 and 14 days. Scale bars: 500 µm in (**A**) and 100 µm in (**B**–**H**) panels.

**Table 1 polymers-13-01568-t001:** Quantitative analysis of fibrillar and nonfibrillar ECM components in control native UM (CTR) and in the stromal substitute of heterotypical substitutes of the human UM cultured in WM, EM, and PM media for 7 and 14 days as determined by picrosirius red, Gomori’s reticulin, alcian blue, PAS histochemistry, and epithelial uroplakins as determined by immunohistochemistry. Results were quantified and compared among groups using the statistical test of Mann–Whitney. Statistically significant values are labeled with asterisks (*).

		Picrosirius red	Gomori’s Reticulin	Alcian Blue	PAS	Uroplakin 2	Uroplakin 3
Mean ± SD	CTR	143.1 ± 24.5	149.8 ± 29.5	136.1 ± 8.1	98.3 ± 38	146 ± 43.9	209.3 ± 24.6
	UM-WM 7D	30.9 ± 2.9	63.7 ± 19.2	108.9 ± 8.5	82.1 ± 35.5	55.1 ± 40.5	27.7 ± 6.5
	UM-WM 14D	89 ± 21.9	65.8 ± 50	109.2 ± 17.6	84.9 ± 15.4	70.5 ± 44.2	31.1 ± 12.4
	UM-EM 7D	46.7 ± 8.3	76.1 ± 32	90.5 ± 10.8	96.8 ± 9.1	59.1 ± 53.7	31.3 ± 14
	UM-EM 14D	60.2 ± 15.2	78.8 ± 14	97.5 ± 6.8	94.4 ± 7	73.9 ± 62.1	32.6 ± 7.9
	UM-PM 7D	46.6 ± 6.9	74.7 ± 7.2	87 ± 5.5	56 ± 13.7	57.9 ± 24.6	34.8 ± 12.8
	UM-PM 14D	56.1 ± 10.3	60.8 ± 18.2	88.4 ± 14.5	52.8 ± 7.1	81.1 ± 29.2	33.2 ± 13
Statistical *p* value	CTR vs. UM-WM 7D	0.00001 *	0.00001 *	0.00002 *	0.31500	0.00032 *	0.00001 *
	CTR vs. UM-WM 14D	0.00013 *	0.00150 *	0.00150 *	0.57874	0.00150 *	0.00001 *
	CTR vs. UM-EM 7D	0.00001 *	0.00001 *	0.00001 *	0.68421	0.00288 *	0.00001 *
	CTR vs. UM-EM 14D	0.00001 *	0.00001 *	0.00001 *	0.79594	0.02323 *	0.00001 *
	CTR vs. UM-PM 7D	0.00001 *	0.00001 *	0.00001 *	0.00288 *	0.00002 *	0.00001 *
	CTR vs. UM-PM 14D	0.00001 *	0.00001 *	0.00001 *	0.00032 *	0.00105 *	0.00001 *
	UM-WM 7D vs. UM-WM 14D	0.00001 *	0.52885	0.68421	0.27986	0.24745	0.68421
	UM-EM 7D vs. UM-EM 14D	0.03546 *	0.97051	0.10512	0.35268	0.91180	0.48125
	UM-PM 7D vs. UM-PM 14D	0.05243	0.08921	0.19032	0.31500	0.08921	0.73936
	UM-WM 7D vs. UM-EM 7D	0.00021 *	0.14314	0.00073 *	0.16549	0.73936	0.79594
	UM-WM 7D vs. UM-PM 7D	0.00004 *	0.07526	0.00001 *	0.06301	0.63053	0.27986
	UM-EM 7D vs. UM-PM 7D	0.97051	0.85343	0.31500	0.00001*	0.85343	0.52885
	UM-WM 14D vs. UM-EM 14D	0.00520 *	0.16549	0.08921	0.07526	0.91180	0.73936
	UM-WM 14D vs. UM-PM 14D	0.00150 *	0.48125	0.02323 *	0.00021 *	0.43587	0.68421
	UM-EM 14D vs. UM-PM 14D	0.68421	0.05243	0.24745	0.00001 *	0.73936	0.97051

**Table 2 polymers-13-01568-t002:** Semiquantitative analysis of relevant epithelial markers in control native UM (CTR) and in the stromal substitute of heterotypical substitutes of the human UM cultured in WM, EM, and PM media for 7 and 14 days as determined by immunofluorescence. Results were assessed as strong positive signal (+++), positive signal (++), mild signal (+), or negative (−).

	Pancytokeratin	Cytokeratin 13	Cytokeratin 7	Cytokeratin 8	Desmoplakin	Tight Junction Protein-1
CTR	+++	+++	+++	+++	+++	+++
UM-WM 7D	++	−	−	+	+	−
UM-WM 14D	++	−	−	+	+	−
UM-EM 7D	+	−	−	−	+	++
UM-EM 14D	+	+	+	+	+	++
UM-PM 7D	++	−	−	++	+	−
UM-PM 14D	++	−	−	+	++	−

## Data Availability

The data presented in this study are available on request from the corresponding author.

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
