# Peer review of "Biofabrication of a Tubular Model of Human Urothelial Mucosa Using Human Wharton Jelly Mesenchymal Stromal Cells"

_polymers, 2021, doi:10.3390/polym13101568_

Round 1

Reviewer 1 Report

Manuscript offered by Garzon et al has been describing the use of human Wharton Jelly Mesenchymal Stromal Cells (here referred as stem cells only, please motivate) as an alternative source of cells able to generate a substitute of the urothelial epithelial layer.

No cell product characterization has been offered to support quality and potency of the initial product (as recommended in several papers, including Dominici et al, Cytotherapy 2006). Such evidence should be offered, at least as supplementary material, if not briefly included in results. The cell isolation approach requires a particularly long digestion time (6hrs) in comparison with classical 1-2hrs incubation. Please motivate. Is the product characteristic the same as the other reports? Cell viability?

The authors correctly mentioned some cellular properties that attracted researcher attention on WJ-MSC: multipotency (so far limited to functional maturation into osteo-/chondro-/adipo-cells; unlimited accessibility and relatively easily/efficient isolation; high rate of proliferation (despite recent evidence suggest significant changes and decrease in growing efficiency after 4 passages, thus it would be critical to state which cells has been included in the current study); low immunological rejection rates (actually MSC have been described no immunogenic and even immune-modulatory in efficacy). Thus, why the authors did not find instrumental to evaluate lymphocyte infiltration upon implantation is not clear. Evaluation in vivo is critical, as much as functional evaluation.

Indeed, the authors reported WJ-MSC capable to differentiate into urothelial cell lineage, but only partially. Such limited efficiency has not been properly described or evaluated.

The present work has elegantly described morphological evaluation, with limited characterization of the final product. The main function of urothelium (containment of water and urine-like solution) should be relative easy to perform. The study has been limitedly focused on immunohistochemical analysis. No further supportive techniques have been included in support or discredit of microscopical evaluations.

Finally, three types of medium formulation have been included in the study: classical medium with or without EGF (classical mitogen), and medium previously preconditioned. Additional analysis and control should be included to proper evaluate effects and efficiency in epithelial maturation.

MSC have been initially described capable to mature into all three germ layers. But later corrected and described limited in their differentiation potential. Few or null exhaustive description and proof of mesenchymal-to-epithelial transition has been proved so far (while the opposite process have been largely described). Preclinical experiments, carried out and supporting clinical infusions where maturation into epithelial cells was required, were initially interpreted as direct proof of MSC “transdifferentiation” into epithelial cells, suddenly “corrected” by fusion event occurred between donor cells and host hepatocytes.

The current version requires critical revision and editing, starting from the title. As the authors stated (lines 442-444) “Although some preliminary signs of urothelial differentiation were found, most induced HWJSC significantly differed from control urothelium from a morphological and a functional levels.”.  Such statement elegantly summarizes novelty and limits of the current study.

Minor details:

Introduction is particularly long and sometimes redundant. Considerations and description, as the section between lines 73-87, would probably fit in the Discussion rather than in Intro section. Results description can benefit from editing and revision (shorten)

Author Response

Reviewer 1.

Manuscript offered by Garzon et al has been describing the use of human Wharton Jelly Mesenchymal Stromal Cells (here referred to as stem cells only, please motivate) as an alternative source of cells able to generate a substitute of the urothelial epithelial layer.

Author´s Response (AU): Following the reviewer’s suggestion, the manuscript has been modified and these cells are now referred to as human Wharton Jelly Mesenchymal Stromal Cells.

No cell product characterization has been offered to support quality and potency of the initial product (as recommended in several papers, including Dominici et al, Cytotherapy 2006). Such evidence should be offered, at least as supplementary material, if not briefly included in results.

AU: This is an important observation. In the present work, human Wharton Jelly Mesenchymal Stromal Cell cultures were characterized according to Dominici’s criteria, and we demonstrated that cells used for the generation of the different tissue substitutes were positive for CD90, CD105 and CD73 markers and negative for CD45. These results have been briefly described in the results section of the revised manuscript, as suggested (page 6, lines 232-235).

The cell isolation approach requires a particularly long digestion time (6hrs) in comparison with classical 1-2hrs incubation. Please motivate. Is the product characteristic the same as the other reports? Cell viability?

AU: The reviewer is right, and we agree that most tissues are completely digested after an enzymatic digestion of 1-2 hours. However, our previous experience demonstrated that some large-size biopsies of very dense tissues may require up to 6 hours of incubation. Therefore, we established a digestion protocol in which tissues are examined every 30 minutes of incubation, and cells are harvested at the moment the tissue becomes completely digested (typically, after 1-2h, but up to 6 hours for big biopsies). This protocol has been clarified and the sentence in the methods section has been reworded for clarity (page 2, lines 93-95).

Regarding cell viability, our previous studies revealed that this procedure is able to generate HWJSC cultures with a viability of 89.35% (Garzón et al., 2012).

  • Garzón, I. et al. (2012) ‘Evaluation of the cell viability of human Wharton’s jelly stem cells for use in cell therapy’, Tissue Engineering. Part C, Methods, 18(6), pp. 408–419. doi: 10.1089/ten.TEC.2011.0508.

The authors correctly mentioned some cellular properties that attracted researcher attention on WJ-MSC: multipotency (so far limited to functional maturation into osteo-/chondro-/adipo-cells; unlimited accessibility and relatively easily/efficient isolation; high rate of proliferation (despite recent evidence suggest significant changes and decrease in growing efficiency after 4 passages, thus it would be critical to state which cells has been included in the current study); low immunological rejection rates (actually MSC have been described no immunogenic and even immune-modulatory in efficacy). Thus, why the authors did not find instrumental to evaluate lymphocyte infiltration upon implantation is not clear. Evaluation in vivo is critical, as much as functional evaluation.

AU: We agree with the reviewer that in vivo evaluation is very important step for tissue characterization. In this regard, previous studies carried out by our research group demonstrated that bioartificial tissues generated with the same fibrin-agarose biomaterial had excellent in vivo biocompatibility once grafted in animal models (Campos et al., 2020) (Garzón et al., 2020) (Chato-Astrain et al., 2018). These results allowed us to use this biomaterial in two different clinical trials in humans, and fibrin-agarose based tissues were grafted in patients with severe corneal damage (Rico-Sánchez et al., 2019) and extensive skin burns (Egea-Guerrero et al., 2019) with good results. It is true that the urothelial mucosa substitutes generated in the present work have not evaluated in vivo, and future works should demonstrate the in vivo biocompatibility of these novel products.

This relevant information has been included in the discussion of the revised manuscript and the need of performing in vivo assays has been considered among the limitations of the present study (page 14, lines 534 and page 15, lines 536-539 and 549-550).

  • Campos, F. et al. (2020) ‘Evaluation of Fibrin-Agarose Tissue-Like Hydrogels Biocompatibility for Tissue Engineering Applications’, Frontiers in Bioengineering and Biotechnology, 8, p. 596. doi: 10.3389/fbioe.2020.00596.
  • Garzón, I. et al. (2020) ‘Long-Term in vivo Evaluation of Orthotypical and Heterotypical Bioengineered Human Corneas’, Frontiers in Bioengineering and Biotechnology, 8, p. 681. doi: 10.3389/fbioe.2020.00681.
  • Chato-Astrain, J. et al. (2018) ‘In vivo Evaluation of Nanostructured Fibrin-Agarose Hydrogels With Mesenchymal Stem Cells for Peripheral Nerve Repair’, Frontiers in Cellular Neuroscience, 12, p. 501. doi: 10.3389/fncel.2018.00501.
  • Rico-Sánchez, L. et al. (2019) ‘Successful development and clinical translation of a novel anterior lamellar artificial cornea’, Journal of Tissue Engineering and Regenerative Medicine, 13(12), pp. 2142–2154. doi: 10.1002/term.2951.
  • Egea-Guerrero, J. J. et al. (2019) ‘Transplant of Tissue-Engineered Artificial Autologous Human Skin in Andalusia: An Example of Coordination and Institutional Collaboration’, Transplantation Proceedings, 51(9), pp. 3047–3050. doi: 10.1016/j.transproceed.2019.08.014.

Indeed, the authors reported WJ-MSC capable of differentiating into urothelial cell lineage, but only partially. Such limited efficiency has not been properly described or evaluated.

AU: As the reviewer states, our results show that the differentiation potential of WJ-MSC was partial. In fact, these cells were able to form a multilayered stratum able to express some components of the human urothelium when specific differentiation media were used. These findings have been emphasized in the manuscript discussion and clearly shown in the conclusion of the revised paper, and the need of evaluating the biological processes related to this partial differentiation potential has been considered as a limitation of the study (page 2, lines 72, page 12, lines 420-421 and 428, page 14, line 507 and 520 and page 15, lines 542, 548 and 549).

The present work has elegantly described morphological evaluation, with limited characterization of the final product. The main function of urothelium (containment of water and urine-like solution) should be relative easy to perform. The study has been limited focused on immunohistochemical analysis. No further supportive techniques have been included in support or discredit of microscopical evaluations.

AU: As the reviewer states, this work is focused on a histological and histochemical characterization of the different tissue substitutes generated by tissue engineering, and a functional characterization should be carried out. Although we have some indirect evidence of tissue function derived from the histological and molecular analyses carried out so far, future studies should evaluate tissue functionality including water containment. Unfortunately, the current situation does not allow us to perform these experiments at the present time. However, we have included this fact as a limitation of the study and we suggest the need of performing thorough functional analysis in future studies (page 14, lines 531-534 and page 15, line 539).

Finally, three types of medium formulation have been included in the study: classical medium with or without EGF (classical mitogen), and medium previously preconditioned. Additional analysis and control should be included to proper evaluate effects and efficiency in epithelial maturation.

AU: This is a good idea that has been considered in the discussion section. In fact, the need of performing these analyses has been referred to as a limitation of the study (page 13, lines 439-441 and page 14, lines 529-531).

MSC have been initially described capable to mature into all three germ layers. But later corrected and described limited in their differentiation potential. Few or null exhaustive description and proof of mesenchymal-to-epithelial transition has been proved so far (while the opposite process have been largely described). Preclinical experiments, carried out and supporting clinical infusions where maturation into epithelial cells was required, were initially interpreted as direct proof of MSC “transdifferentiation” into epithelial cells, suddenly “corrected” by fusion event occurred between donor cells and host hepatocytes.

AU: This is an interesting observation. As the reviewer states, the differentiative potential of human MSC is controversial and needs further characterization.

In this regard, we previously demonstrated that human MSC were able to differentiate into several types of cells such as skin keratinocytes and cornea epithelial cells (Alaminos et al., 2010; Garzón et al., 2013, 2014, 2015, 2020; Martin-Piedra et al., 2019) both ex vivo and in vivo, although the efficiency of the differentiative process was higher in vivo (reviewed in Garzon et al., 2020). However, it is true that several reports suggest that their differentiation potential could not be very high and a process of cell fusion could play a crucial role on their functional effect in vivo (Alvarez-Dolado et al., 2003; Charbord, 2010).

This important issue has been incorporated to the revised manuscript (page 12, lines 411-412).

  • Alaminos, M. et al. (2010) ‘Transdifferentiation potentiality of human Wharton’s jelly stem cells towards vascular endothelial cells’, Journal of Cellular Physiology, 223(3), pp. 640–647. doi: 10.1002/jcp.22062.
  • Garzón, I. et al. (2013) ‘Wharton’s jelly stem cells: a novel cell source for oral mucosa and skin epithelia regeneration’, Stem Cells Translational Medicine, 2(8), pp. 625–632. doi: 10.5966/sctm.2012-0157.
  • Garzón, I. et al. (2014) ‘Generation of a biomimetic human artificial cornea model using Wharton’s jelly mesenchymal stem cells’, Investigative Ophthalmology & Visual Science, 55(7), pp. 4073–4083. doi: 10.1167/iovs.14-14304.
  • Garzón, I. et al. (2020) ‘Long-Term in vivo Evaluation of Orthotypical and Heterotypical Bioengineered Human Corneas’, Frontiers in Bioengineering and Biotechnology, 8, p. 681. doi: 10.3389/fbioe.2020.00681.
  • Garzón, I., Martin-Piedra, M. A. and Alaminos, M. (2015) ‘Human Dental Pulp Stem Cells. A promising epithelial-like cell source’, Medical Hypotheses, 84(5), pp. 516–517. doi: 10.1016/j.mehy.2015.02.020.
  • Martin-Piedra, M. A. et al. (2019) ‘Effective use of mesenchymal stem cells in human skin substitutes generated by tissue engineering’, European Cells & Materials, 37, pp. 233–249. doi: 10.22203/eCM.v037a14.
  • Garzon, I. et al. (2020) ‘Expanded Differentiation Capability of Human Wharton’s Jelly Stem Cells Toward Pluripotency: A Systematic Review’, Tissue Engineering. Part B, Reviews. doi: 10.1089/ten.TEB.2019.0257.
  • Alvarez-Dolado, M. et al. (2003) ‘Fusion of bone-marrow-derived cells with Purkinje neurons, cardiomyocytes and hepatocytes’, Nature, 425(6961), pp. 968–973. doi: 10.1038/nature02069.
  • Charbord, P. (2010) ‘Bone marrow mesenchymal stem cells: historical overview and concepts’, Human Gene Therapy, 21(9), pp. 1045–1056. doi: 10.1089/hum.2010.115.

The current version requires critical revision and editing, starting from the title. As the authors stated (lines 442-444) “Although some preliminary signs of urothelial differentiation were found, most induced HWJSC significantly differed from control urothelium from a morphological and a functional levels.”.  Such statement elegantly summarizes novelty and limits of the current study.

AU: The whole manuscript has been carefully revised and edited, as suggested. The idea that differentiation was not complete has been highlighted all over the manuscript and has been incorporated to the conclusion paragraph of the revised manuscript to highlight its relevance (page 15, lines 548-549).

Minor details:

Introduction is particularly long and sometimes redundant. Considerations and description, as the section between lines 73-87, would probably fit in the Discussion rather than in Intro section. Results description can benefit from editing and revision (shorten).

AU: The referred paragraph has been transferred to the discussion section and the results section has been revised, edited and shortened as suggested.

Reviewer 2 Report

In the work by Ingrid Garzón et al., authors have been able to generate bioartificial UM able to resemble native urothelial mucosa in terms of morphology, histology, and functional similarities with the native tissue. The topic is very interesting and actual, unfortunately the manuscript does not have a significant impact, indeed it is not particularly addressed of experiments. Nevertheless, it seems well structured and easy to read. As a tip  I would suggest reducing the discussion part by making shorter .

Author Response

Reviewer 2.

In the work by Ingrid Garzón et al., authors have been able to generate bioartificial UM able to resemble native urothelial mucosa in terms of morphology, histology, and functional similarities with the native tissue. The topic is very interesting and actual, unfortunately the manuscript does not have a significant impact, indeed it is not particularly addressed of experiments. Nevertheless, it seems well structured and easy to read. As a tip  I would suggest reducing the discussion part by making shorter.

Author´s Response (AU): Thank you very much for your kind comments. The whole manuscript text has been edited and shortened, as suggested.

Reviewer 3 Report

 Surgical reconstruction of  urothelial mucosa (UM) remains one of the most challenging procedures in urology and is frequently associated with complications, restenosis and poor quality of life for the affected individual.  Tissue engineering using different cell types and tissue scaffolds offers a promising alternative for tissue repair and replacement. Most strategies to engineer the UM comprise two components: a scaffold, which provides structure, and cells, which provide a barrier from transported fluids. In the paper are presented the results of a study by bioartificial UM based on nanostructured fibrin-agarose biomaterials containing human fibroblasts isolated from ureter stroma were used, and human Wharton jelly stem cells (HWJSC) - are used to generate an epithelial-like layer on top.

The work is well structured, written in good language. But there are some inaccuracies and questions:

  1. The methods indicate how much weight (500 ml) and how long (3 minute) the load is kept on the layer containing the cells. But it is not described why this weight and time was chosen?

  1. How viable are the cells in the hydrogel before and after dehydration?

  1. As far as the tubular structure formed by stitching will be complete, its integrity is broken. It is not clear from the text how overgrowing of this defect is possible.

  1. In the methods section, the dye alcian blue is presented and the results are given for the staining of Toluidine blue, these are different dyes, although they stain proteglycans. Please clarify

  1. Phrase «Briefly, staining intensity was scored as strong positive signal (+++), positive signal 198 (++), mild signal (+), slightly positive signal (±) or negative (–)». It is not clear by what criteria the gradations are divided, there are image analysis programs for example ImageJ, maybe the authors should present numerical expressions. References 29 and 30 are not entirely clear, the authors refer to their own articles - the same phrase is presented there without clarification on what parameters for the presented types of color measurement.

  1. Figure 2А и 2 G and 4D, 3H scale bare are different from other. Why it is?

  1. Lines 242-243. «Then, we found that the UM were composed by a fibrin- agarose biomaterial containing scattered elongated, spindle shaped cells that was compatible with a stromal substitute, and a cell layer resembling an epithelium on top.» The presented figures is not demonstrate  cells in fibrin- agarose layers (for example fig 2E). But authors describe what they are. Either they are poorly stained, or the cells have destroyed.

  1. Cells in the lower epithelial layers are vacuolated (UM-EM). What is it? Is it a staining artifact, or is the cells stressed, maybe autophagy?

  1. There are also typos in the text, for example:

Line 330 urothe-lial

Author Response

Reviewer 3.

Surgical reconstruction of urothelial mucosa (UM) remains one of the most challenging procedures in urology and is frequently associated with complications, restenosis and poor quality of life for the affected individual.  Tissue engineering using different cell types and tissue scaffolds offers a promising alternative for tissue repair and replacement. Most strategies to engineer the UM comprise two components: a scaffold, which provides structure, and cells, which provide a barrier from transported fluids. In the paper are presented the results of a study by bioartificial UM based on nanostructured fibrin-agarose biomaterials containing human fibroblasts isolated from ureter stroma were used, and human Wharton jelly stem cells (HWJSC) - are used to generate an epithelial-like layer on top.

The work is well structured, written in good language. But there are some inaccuracies and questions:

  1. The methods indicate how much weight (500 ml) and how long (3 minute) the load is kept on the layer containing the cells. But it is not described why this weight and time was chosen?

Author´s Response (AU): Thank you for your kind comments. Regarding nanostructuration, this protocol was previously optimized and standardized by our research group for different types of biomaterials based on fibrin and agarose. As shown by Scionti et al. (2014), the specific conditions used for the bioartificial UM -500g and 3min- were the most appropriate for this type of biomaterial and were able to generate a dense hydrogel with significantly improved biomechanical properties. This specific protocol was able to generate a biomaterial with adequate mechanical behavior, allowing the efficient biofabrication 3D tubular models.

This has been clarified in the revised manuscript methods section (page 4, lines 144-146).

  • Scionti, G. et al. (2014) ‘Effect of the hydration on the biomechanical properties in a fibrin-agarose tissue-like model’, Journal of Biomedical Materials Research. Part A, 102(8), pp. 2573–2582. doi: 10.1002/jbm.a.34929.

  1. How viable are the cells in the hydrogel before and after dehydration?

AU: As stated above, nanostructuration protocols were previously standardized by our research group for fibrin-agarose biomaterials containing human cells, and we optimized the protocols that preserved cell viability with the highest efficiency. In this regard, we previously demonstrated that fibrin-agarose biomaterials containing human cells subjected to the same nanostructuration protocol used in the present work contained fully viable cells with high expression of cell proliferation markers such as PCNA and Ki-67, negative expression of Caspase 7 as a maker of cell death (Chato-Astrain et al., 2020) and active synthesis of relevant tissue components (Carriel et al., 2017), suggesting that cells were viable. In fact, demonstration that nanostructured fibrin-agarose hydrogels contained viable cells allowed us to apply this method for the generation of bioartificial human corneas and skin that obtained approval by the Spanish Medicines Agency and are currently used clinically in patients with corneal blindness and severe skin burns (Rico-Sánchez et al., 2019) (Egea-Guerrero et al., 2019).

  • Chato-Astrain, J. et al. (2020) ‘Generation of a novel human dermal substitute functionalized with antibiotic-loaded nanostructured lipid carriers (NLCs) with antimicrobial properties for tissue engineering’, Journal of Nanobiotechnology, 18. doi: 10.1186/s12951-020-00732-0.
  • Carriel, V. et al. (2017) ‘In vitro characterization of a nanostructured fibrin agarose bio-artificial nerve substitute’, Journal of Tissue Engineering and Regenerative Medicine, 11(5), pp. 1412–1426. doi: 10.1002/term.2039.
  • Rico-Sánchez, L. et al. (2019) ‘Successful development and clinical translation of a novel anterior lamellar artificial cornea’, Journal of Tissue Engineering and Regenerative Medicine, 13(12), pp. 2142–2154. doi: 10.1002/term.2951.
  • Egea-Guerrero, J. J. et al. (2019) ‘Transplant of Tissue-Engineered Artificial Autologous Human Skin in Andalusia: An Example of Coordination and Institutional Collaboration’, Transplantation Proceedings, 51(9), pp. 3047–3050. doi: 10.1016/j.transproceed.2019.08.014.

  1. As far as the tubular structure formed by stitching will be complete, its integrity is broken. It is not clear from the text how overgrowing of this defect is possible.

AU: This is an interesting point. In the present work, we have described a biofabrication method based on nanostructuration protocols associated to tissue tubularization and suturing. Although this method succeeded in generating hollow tubular structures, the problems associated to stitching need further research. This issue has been considered in the discussion section of the revised manuscript (page 12, lines 402-404).

  1. In the methods section, the dye alcian blue is presented and the results are given for the staining of Toluidine blue, these are different dyes, although they stain proteglycans. Please clarify.

AU: In the present work, we used semithin tissue sections embedded in epoxy resin and stained with toluidine blue to evaluate tissue structure and morphology. These results are shown in Figure 2 and described in the first paragraph of section 2.3 of the methods section. Then, several ECM components were stained and evaluated on tissue sections embedded in paraffin using several histochemical and immunohistochemical methods, including alcian blue for proteoglycans detection (shown in Figure 4 and described in the second paragraph of section 2.3).

Description of the toluidine blue staining protocol has been improved in the revised manuscript (page 5, lines 172-176).

  1. Phrase «Briefly, staining intensity was scored as strong positive signal (+++), positive signal 198 (++), mild signal (+), slightly positive signal (±) or negative (–)». It is not clear by what criteria the gradations are divided, there are image analysis programs for example ImageJ, maybe the authors should present numerical expressions. References 29 and 30 are not entirely clear, the authors refer to their own articles - the same phrase is presented there without clarification on what parameters for the presented types of color measurement.

AU: Numerical quantification of the results is a very good suggestion. Therefore, we have used the ImageJ software to quantify the signal intensity of the histochemical and immunohistochemical analyses carried out in the present work, as suggested. These analyses allowed us to perform statistical comparisons among study groups. These results have been incorporated to Table 1 of the revised manuscript and the analysis method has been included in the methods section (page 5, lines 219-225 and page 6, lines 226-229).

Regarding the immunofluorescence analyses performed for cytokeratins and cell-cell junctions, intensity results cannot be quantified, since fluorescence intensity is not proportional to the level of protein expression. In these cases, a semiquantitative analysis protocol was applied instead. This procedure has been explained in the methods section and more adequate references, including some from other research groups, were incorporated to the manuscript (page 5, lines 214-219).

  1. Figure 2А и 2 G and 4D, 3H scale bare are different from other. Why it is?

AU: All scale bars have been thoroughly revised and checked. The small differences among panels are due to actual magnification differences.

  1. Lines 242-243. «Then, we found that the UM were composed by a fibrin- agarose biomaterial containing scattered elongated, spindle shaped cells that was compatible with a stromal substitute, and a cell layer resembling an epithelium on top.» The presented figures is not demonstrate cells in fibrin- agarose layers (for example fig 2E). But authors describe what they are. Either they are poorly stained, or the cells have destroyed.

AU: Figure 2 has been modified and new images have been taken from each sample stained with hematoxylin-eosin. The new images show stromal cells in the fibrin-agarose biomaterial, and some of these cells have been labeled with arrows in Figure 2.

  1. Cells in the lower epithelial layers are vacuolated (UM-EM). What is it? Is it a staining artifact, or is the cells stressed, maybe autophagy?

AU: As stated above, Figure 2 has been modified and new images with higher quality have been incorporated to the Figure. These images are compatible with normal cells at the epithelial-like layer.

  1. There are also typos in the text, for example: Line 330 urothe-lial

AU: This manuscript has been thoroughly revised and typographical errors have been corrected.